# Length of hospital stay and its associated factors among women who gave birth by cesarean section in general hospitals of Sidama region, Ethiopia

**Amelo Bolka**[ID]*, **Zerihun Weldekidan**

School of Public Health, Yirgalem Hospital Medical College, Yirgalem, Ethiopia

* amelobolka@gmail.com

## Abstract

Despite rising cesarean section (CS) rates in Ethiopia, evidence on determinants of postoperative length of hospital stay (LoS) remains scarce, particularly for rural general hospitals handling most deliveries. This study was aimed at assessing the length of hospital stay and its associated factors among women who undergo cesarean section in general hospitals of the Sidama region. An institution-based cross-sectional study was conducted among 505 post-CS mothers from 1 January to 20 February 2024. A multistage sampling method was followed to select the study respondents. Data was collected using a structured and pretested, interviewer-administered questionnaire. Data was collected using the Kobo Toolbox system and exported to Stata version 14.0 for management and analysis. Factors associated with the length of hospital stay were determined using a Poisson regression model. The factors associated with the outcome variable were identified using the adjusted risk ratio (ARR). Statistical significance was set at a p-value of less than 0.05. The median LoS post-CS was 4 days (interquartile range: 3–4). Significant predictors of prolonged LoS included maternal age (ARR = 1.014, 95% CI: 1.004–1.024), neonatal intensive care unit (NICU) admission (ARR = 1.31, 95% CI: 1.16–1.46), surgical site infection (ARR = 2.39, 95% CI: 1.88–3.04), and low postoperative hemoglobin (ARR = 0.94, 95% CI: 0.92–0.97). The median hospital stay after cesarean delivery in general hospitals of Sidama region was 4 days. Prolonged stays were associated with maternal age, NICU admission, surgical site infection, and low post-op hemoglobin. Targeting high-risk mothers with enhanced monitoring and wound care—alongside NICU-maternity service integration and safety-conscious discharge protocols—is recommended to accelerate recovery.

**Data availability statement:** All relevant data are within the paper and its Supporting Information files.

**Funding:** The author(s) received no specific funding for this work.

**Competing interests:** The authors have declared that no competing interests exist.

## Introduction

Cesarean section (CS) is a surgical procedure in which the baby is delivered through incisions made in the mother's abdomen and uterus [1]. Although CS is a lifesaving intervention in certain obstetric conditions, it is associated with various short- and long-term adverse health outcomes [2]. These include an increased risk of uterine rupture, abnormal placentation, ectopic pregnancy, stillbirth, preterm birth, surgical site infections, intraoperative hemorrhage, maternal death, febrile morbidity, and extended incision healing time [3,4]. Despite these risks, the global rate of CS continues to rise, with the World Health Organization (WHO) reporting that 21% of all births worldwide are delivered via CS, equating to approximately one in five deliveries [5].

Length of hospital stay (LoS) following CS is an essential indicator of healthcare quality, efficiency, and patient safety [6]. It is defined as the duration a mother spends in a postnatal ward after undergoing CS [5]. The length of hospital stay following CS varies widely across different healthcare settings, influenced by maternal, neonatal, surgical, and institutional factors [7]. Studies indicate that the average LoS after CS ranges from 2 to 7 days in most settings, depending on post-operative recovery, complication rates, and healthcare system efficiency [7–11]. Recent advancements in surgical care, such as Enhanced Recovery After Surgery (ERAS) protocols, have been implemented to reduce LoS, minimize healthcare costs, improve maternal outcomes, enhance maternal satisfaction, and accelerate recovery while mitigating the risk of complications [12].

The LoS after cesarean CS is influenced by maternal, neonatal, surgical, and institutional factors. Maternal factors include pre-existing conditions including hypertension or diabetes, CS type (elective vs. emergency), and parity, which can prolong recovery due to complications [10,13,14]. Neonatal factors like preterm birth, low birth weight, and NICU admission often extend hospitalization for both mother and newborn, requiring integrated care [15]. Surgical factors, such as anesthesia type, technique, and post-operative complications, also impact LoS [16]. Institutional factors, including postnatal care protocols, discharge policies, nursing quality, and health insurance, further influence LoS, with structured care enabling early discharge and resource gaps causing delays [17].

In Ethiopia, the length of post-cesarean hospital stays varies widely due to disparities in healthcare infrastructure, regional economic differences, and inconsistent access to maternal health services [18–20]. Prolonged hospitalization is often a consequence of clinical complications such as surgical site infections or delayed wound healing, compounded by inadequate postnatal follow-up and transportation challenges, particularly in rural areas [21,22]). Extended stays not only increase the risk of hospital-acquired infections and exacerbate hospital overcrowding but also place financial strain on both families and the healthcare system [23,24]. Conversely, premature discharge—often driven by resource constraints or cultural factors—can elevate the risk of post-surgical complications and maternal mortality [25,26]. This delicate balance between ensuring adequate recovery and optimizing hospital resources highlights the need for a deeper understanding of the context-specific determinants of hospital stay duration, ensuring both maternal safety and healthcare system efficiency [27].

Despite the rising number of cesarean deliveries in Ethiopia, research on the factors influencing hospital stay duration among post-CS women remains limited. While some studies have been conducted in urban hospitals, there is a scarcity of data from rural general hospitals, where the majority of deliveries take place. The Sidama region, with its distinct socio-demographic characteristics and healthcare infrastructure, necessitates localized evidence to inform policy and practice. Therefore, this study aims to assess the length of hospital stay and its associated factors among women who undergo cesarean section in general hospitals of the Sidama region.

## Methods and materials

### Ethics statement

This study adhered to the ethical principles of the Declaration of Helsinki for medical research involving human subjects. Ethical approval was obtained from the Institutional Review Board (IRB) of Yirgalem Hospital Medical College (Protocol Number-IRB/015/23, Date-02/11/2023). Written informed consent was obtained from study participants after explaining the study's purpose, and participant data were anonymized using coded identifiers to ensure confidentiality.

### Study area and period

The study was conducted in the Sidama Region, one of Ethiopia's twelve regional states, located 273 kilometers south of Addis Ababa. The region is administratively divided into four zones, thirty rural districts, six town administrations, and one city administration. According to the 2007 Ethiopia Central Statistical Agency report estimation, the region's population was 4,748,639, with 2,398,063 females and 2,350,576 males. There were 160,809 pregnant women in the region at the time. The region has 553 health posts, 141 health centers, 23 primary hospitals, 8 general hospitals, and one tertiary hospital. All general and tertiary hospitals provide cesarean delivery services free of charge. Based on 2024 reports from the general hospitals, 3,082 women gave birth via cesarean section. This study was conducted from 1 January to 20 February 2024.

### Study population and their eligibility criteria

Postpartum mothers in the Sidama Region who delivered via cesarean section in general hospitals during the study period were included, while those with mental illness or who gave birth before 28 weeks of gestation were excluded.

### Sample size determination

The sample size was calculated using the formula $n = ((Z_{\alpha/2})^2 * (SD)^2 / (E)^2)$ to estimate the length of hospital stay. The standard deviation (SD) of 2 was obtained from a previous study conducted in Sudan [9]. The precision ($E^2 = 0.028$) was determined by multiplying the Z-score corresponding to the desired confidence level by the standard error of the mean. The assumptions considered were a 95% confidence level, 80% power, and a 10% non-response rate. Based on these parameters, the initial sample size was computed as 604. Given that the total source population consisted of 3,082 mothers who had given birth by cesarean section in general hospitals within the region, the sample size was adjusted accordingly. After applying the correction, the final sample size used for this study was 505.

### Sampling procedure

A multistage sampling method was employed in this study. First, three general hospitals (Adare, Yirgalem, and Bona) were selected using a simple random sampling technique. A systematic sampling technique was then applied to select a representative sample of mothers from each chosen hospital. The sample size was allocated proportionally based on the post-CS cases recorded in the previous two months, as reviewed from postnatal registration logbooks in each hospital. Every post-CS mother was included until the desired sample size was reached, as the sampling interval was determined to be one, based on the hospitals' previous two-month post-CS registration (K = 558/505 ≈ 1).

### Study variables

The outcome variable of interest was length of hospital stay following cesarean delivery. On other hand fourteen independent variables were considered for the study. These were: (1) age of mother, (2) parity, (3) number of ANC visits, (4) gestational age at birth, (5) APGAR score, (6) postoperative hemoglobin, (7) previous history of abortion, (8) history of chronic medical problems, (9) pre-delivery admission, (10) neonatal history of NICU admissions, (11) current obstetric complications, (12) surgical site infections, (13) anesthesia complications, and (14) cesarean hysterectomy.

### Data collection instruments and procedures

A structured questionnaire was used to collect data on socio-demographic and economic characteristics, maternal health, and obstetric and neonatal-related factors. The questionnaire was developed based on the study objectives and previous literature [7,9,15]. It was carefully designed, pretested, and administered by trained interviewers. Data collection was conducted by a team of six midwife nurses (Bachelor of Science holders) through face-to-face interviews. Postoperative maternal hemoglobin levels, birth weight, APGAR scores, wound infection, and LoS were obtained from medical record reviews. Two Master of Public Health (MPH) degree holders supervised the overall data collection process. The KoboToolbox application, installed on Android devices, was used for data collection. The collected data were automatically submitted to a central server.

### Data quality control

Qualified data collectors and supervisors received comprehensive training on the KoboToolbox system and interview skills. The data collection tool was pre-tested on 5% of the sample, and necessary modifications were made. Rigorous supervision included daily examinations and prompt error corrections. The use of the KoboToolbox system for data collection facilitated logical data entry and maintained data quality. Supervisors verified form completeness, ensuring data integrity throughout the process.

### Data management and analysis

Data were collected using the KoboToolbox system and exported to Stata version 14.0 for management and analysis. Descriptive statistics, including frequency distributions and measures of central tendency and dispersion, were used to summarize the data. Factors associated with length of hospital stay were analyzed using a Poisson regression model. Prior to analysis, model assumptions (including equidispersion) were checked and satisfied. In bivariable Poisson regression analysis, variables with a p-value < 0.25 were considered candidate and were subsequently included in the multivariable Poisson regression model to assess their association with the outcome variable. The multivariable analysis identified statistically significant factors using adjusted risk ratios (ARR) with 95% confidence intervals (CI). A p-value threshold of < 0.05 was set for statistical significance.

## Results

### Socio demographic characteristics of the study participants

A total of 497 mothers participated in this study, yielding a response rate of 98.4%. The mean age of the participants was 27 years (±5.2 standard deviation). The median monthly income was 5,000 Ethiopian Birr ($US 52.63), with an interquartile range (IQR) of 1,600–5,000 Ethiopian Birr. The vast majority of the mothers (96.6%) were married, and about two-thirds (65.8%) identified as followers of the Protestant religion. In terms of education, 78.7% of the participants and 77.9% of their husbands had attended primary school or higher. Regarding occupation, 47.5% of the mothers were housewives, while 39.2.5% of their husbands were farmers. More than half of the participants (56.9%) resided in urban areas (Table 1).

**Table 1. Socio-demographic characteristics of study participants at general hospitals of Sidama region, southern Ethiopia.**

| Variable | Category | Frequency | Percent (%) |
|---|---|---|---|
| Marital status | Married | 480 | 96.6 |
| | Divorced | 17 | 3.4 |
| Religion | Protestant | 327 | 65.8 |
| | Orthodox | 131 | 26.4 |
| | Muslim | 39 | 7.8 |
| Mothers' educational | Not attended formal education | 106 | 21.3 |
| | Primary school and above | 391 | 78.7 |
| Mothers' occupation | Housewife | 236 | 47.5 |
| | Employed | 102 | 20.5 |
| | Private work | 159 | 32.0 |
| Husbands' educational | Not attended formal education | 110 | 22.1 |
| | Primary school and above | 387 | 77.9 |
| Husbands' occupation | Employed | 164 | 33.0 |
| | Private work | 138 | 27.8 |
| | Farmer | 195 | 39.2 |
| Residence | Rural | 214 | 43.1 |
| | Urban | 283 | 56.9 |

## Maternal and obstetric health related characteristics

The median duration of a mother's stay in labor was 9 hours (IQR: 6–14). Slightly less than two-thirds (62.2%) of the mothers had given birth to two or more times to viable fetuses (multipara). Four-fifths (83.3%) of the study participants reported no history of previous abortions. One hundred eighty-nine (38%) mothers had delivered vaginally in their previous pregnancy. The vast majority (89.7%) of mothers visited health institutions for at least one antenatal care. Twenty-five (5%) participants had a history of chronic medical conditions. Seventy-seven (15.5%) of the mothers were admitted before delivery (Table 2).

## Neonatal factors

The majority (89.1%) of the study participants had given birth at term. Four hundred sixty-four (93.4%) mothers delivered babies with normal birth weights (≥ 2500 grams). Twenty-seven (6.2%) of the neonates had low APGAR scores (≤6 scores) at five minutes. One-fifth (19.5%) of the neonates were admitted to the neonatal intensive care unit (NICU) (Table 3).

## Obstetrical factors

The vast majority (90.3%) of the study participants underwent an emergency cesarean section. A lower transverse skin incision was the most common type of CS (92.6%) performed in the included health institutions. Spinal or local anesthesia was administered to four hundred seventy-six (95.8%) mothers during the CS procedure. Five mothers (1%) experienced anesthesia-related complications. One-fifth (20.7%) of the mothers had current obstetric complications. Seven mothers (1.4%) developed surgical site infections. Fourteen mothers (2.8%) underwent cesarean hysterectomy. The mean (± SD) postoperative hemoglobin level of the study participants was 11.24 ± 1.47 g/dl (Table 4).

## Length of hospital stays after cesarean section

The median length of hospital stay after cesarean delivery was 4 days, with an IQR of 3–4 days. The minimum length of stay was 2 days, and the maximum was 17 days. About half (49.9%) of the respondents were discharged within 2–3 days.

**Table 2. Obstetric related characteristics of study participant in general public hospitals of Sidama region, southern Ethiopia.**

| Variable | Frequency | Percent (%) |
|---|---|---|
| Parity | | |
| Primipara (given birth once to a viable fetus) | 158 | 31.8 |
| Multipara (delivered ≥2 viable fetuses) | 309 | 62.2 |
| Grand multipara (delivered ≥5 viable fetuses) | 30 | 6.0 |
| Previous pregnancy mode of delivery | | |
| Vaginal | 189 | 38.0 |
| Cesarean Section | 152 | 30.6 |
| Primi para | 156 | 31.4 |
| Previous abortion | | |
| Yes | 83 | 16.7 |
| No | 414 | 83.3 |
| ANC visit | | |
| Yes | 446 | 89.7 |
| No | 51 | 10.3 |
| Known chronic medical conditions[*] | | |
| Yes | 25 | 5.0 |
| No | 472 | 95.0 |
| Admitted before delivery | | |
| Yes | 77 | 15.5 |
| No | 420 | 84.5 |

[*]The chronic medical diseases considered in this study included chronic hypertension without proteinuria, cardiovascular diseases, pre-gestational diabetes mellitus, asthma, and chronic pulmonary diseases.

**Table 3. Neonatal related characteristics of study participants in general public hospitals of Sidama region, southern Ethiopia.**

| Variable | Frequency | Percent |
|---|---|---|
| Gestational age of neonate at birth | | |
| Preterm | 42 | 8.5 |
| Term | 443 | 89.1 |
| Post term | 12 | 2.4 |
| Birth weight of neonate in gram | | |
| Low birth weight (< 2500 grams) | 33 | 6.6 |
| Normal birth weight (≥ 2500 grams) | 464 | 93.4 |
| APGAR score in 5th minute | | |
| Low APGAR score (≤6) | 27 | 6.2 |
| Normal APGAR score (7–10) | 466 | 93.8 |
| Neonate admitted to NICU | | |
| Yes | 97 | 19.5 |
| No | 400 | 80.5 |

APGAR: Appearance, Pulse, Grimace, Activity, Respiration; NICU: Neonatal Intensive Care Unit.

**Table 4. Obstetric related characteristics of study participants in public general hospitals of Sidama region, southern Ethiopia.**

| Variable (n = 497) | Frequency | Percent (%) |
|---|---|---|
| Types of cesarean section | | |
| Emergency | 449 | 90.3 |
| Elective | 48 | 9.7 |
| Types of skin incision | | |
| Lower transvers | 460 | 92.6 |
| Vertical | 37 | 7.4 |
| Current obstetric problems* | | |
| Yes | 103 | 20.7 |
| No | 394 | 79.3 |
| Surgical site infection | | |
| Yes | 7 | 1.4 |
| No | 490 | 98.6 |
| Type of anesthesia used | | |
| Spinal/ local anesthesia | 476 | 95.8 |
| General anesthesia | 21 | 4.2 |
| Anesthesia complication | | |
| Yes | 5 | 1.0 |
| No | 492 | 99.0 |
| Cesarean hysterectomy | | |
| Yes | 14 | 2.8 |
| No | 483 | 97.2 |

*This study considered the following current obstetric complications: hypertensive disorders of pregnancy, antepartum hemorrhage, gestational diabetes mellitus, premature rupture of membranes, congenital anomalies, neonatal loss, and intrauterine growth restriction.

### Factors associated with length of hospital stay after cesarean section

Fourteen variables showed an association with the length of hospital stay after CS on a binary Poisson regression analysis. In multivariable Poisson regression analysis, age of mother, post-operation hemoglobin level, neonate admission to NICU, and surgical site infection were significantly associated with length of hospital stay after CS ($P < 0.05$).

As women's age increases, the likelihood of staying in the hospital after CS increases by 1.4% compared with their counterparts (ARR = 1.014, 95% CI: 1.004–1.024). Women who had a neonate admitted to the NICU had a 31% higher likelihood of staying in the hospital after CS than their counterparts (ARR = 1.31, 95% CI = 1.16–1.46). Holding all other variables constant, a one-unit increase in surgical site infections was associated with a 2.386-fold increase in staying in the hospital after CS (ARR = 2.39, 95% CI = 1.88–3.04). A one-unit increase in mothers' postoperative hemoglobin level (ARR = 0.94, 95% CI: 0.92–0.97) led to a 0.94-fold decrease in the hospital stay after CS (Table 5).

### Discussion

This study assessed length of hospital stay and its associated factors among women who give birth by cesarean section in general hospitals of Sidama region. The median length of hospital stay after cesarean delivery was 4 days with an IQR of 3–4 days. Factors associated with length of hospital stay after CD included age of mothers, post operation hemoglobin level, neonate admission to NICU and surgical site infection.

This study found that the median LoS following cesarean section in general hospitals of the Sidama region was 4 days (IQR: 3–4 days). Studies in South Ethiopia [13] and Sudan [9] have reported comparable LoS for mothers who underwent

**Table 5. Poisson Regression Analysis result of the Factors Associated with LoS after CS in Sidama region public general hospitals, southern Ethiopia.**

| Variable | Categories | CRR | ARR (95% CI) | P- Value |
|---|---|---|---|---|
| Previous abortion | Yes | 1.13 | 1.01 (0.90, 1.14) | 0.807 |
| | No | 1 | 1 | . |
| Known chronic medical disease | Yes | 1.35 | 1.09 (0.90, 1.33) | 0.360 |
| | No | 1 | 1 | . |
| Admitted before delivery | Yes | 1.47 | 1.09 (0.95, 1.24) | 0.220 |
| | No | 1 | 1 | . |
| Neonate Admitted to NICU | Yes | 1.34 | 1.31 (1.16, 1.46) | < 0.001* |
| | No | 1 | 1 | . |
| Current obstetric problems | Yes | 2.90 | 1.09 (0.97, 1.23) | 0.130 |
| | No | 1 | 1 | . |
| Surgical site infection | Yes | 1.41 | 2.39 (1.88, 3.04) | < 0.001* |
| | No | 1 | 1 | . |
| Anesthetic complications | Yes | 1.49 | 1.42 (0.95, 2.13) | 0.091 |
| | No | 1 | 1 | . |
| Caesarean hysterectomy | Yes | 1.51 | 0.99 (0.76, 1.30) | 0.972 |
| | No | 1 | 1 | . |
| Ureteric injury happens | Yes | 1.21 | 1.53 (0.79, 2.97) | 0.207 |
| | No | 1 | 1 | . |
| Age of respondent | Mean | 1.02 | 1.014 (1.004, 1.024) | 0.006* |
| Parity | Mean | 1.04 | 0.98 (0.94, 1.02) | 0.318 |
| Gestational age in week | Median | 0.98 | 1.01 (0.98, 1.04) | 0.455 |
| APGAR score in 5th minute | Mean | 0.91 | 0.97 (0.94, 1.01) | 0.121 |
| Post operation hemoglobin | Mean | 0.92 | 0.94 (0.92, 0.97) | < 0.001* |

CS. However, studies Eritrea [8], Uganda [28], India [10], Italy [11], and Australia [29] have documented a higher median LoS for mothers who underwent cesarean section. These variations may be attributed to differences in healthcare infrastructure, sample sizes, study settings, and hospital discharge policies. Moreover, the presence of post-surgical complications, comorbidities, the need for additional medical interventions such as blood transfusions, and institutional guidelines on postoperative care could all contribute to discrepancies in LoS across different settings.

The study found a significant association between maternal age and LoS. As mothers' age increases, the likelihood of staying in the hospital after CS increases by 1.4% compared with their counterparts (ARR = 1.014, 95% CI: 1.004–1.024). This finding is consistent with a study conducted in Nepal [30], Czechia [31], low income countries [32], Brazil [33] which also reported extended LoS among older mothers. The prolonged LoS in older women may be attributed to age-related factors such as slower wound healing, altered metabolic responses, higher prevalence of pre-existing comorbidities, and increased risk of postoperative complications requiring extended monitoring and medical intervention.

Postoperative hemoglobin level significantly influenced LoS in this study, with higher levels associated with earlier discharge (ARR = 0.94, 95% CI: 0.91-0.97). This finding is consistent with previous research demonstrating that optimal postoperative hemoglobin levels predict shorter hospitalization durations [34]. The observed association likely reflects several physiological advantages: women with normal-range hemoglobin levels typically experience better postoperative recovery, minimizing the need for extended care. Adequate hemoglobin levels may promote faster healing by improving

tissue oxygenation while reducing the risk of anemia-related complications, both of which contribute to more efficient recovery and earlier discharge eligibility.

Surgical site infection was significantly associated with prolonged LoS, with women who developed SSI staying longer than those who did not (ARR = 2.58, 95% CI: 2.03–3.27). This finding aligns with studies from southern Ethiopia [13], eastern Ethiopia [24], Uganda [28] and Sudan [9]. The extended LoS likely results from delayed wound healing requiring additional treatment, potential systemic complications needing IV antibiotics or reoperation, time for diagnostic confirmation, and mandatory hospitalization until infection resolution. In resource-limited settings, delayed detection and limited antibiotic access further exacerbate these delays.

Neonatal admission to the NICU was also a significant predictor of prolonged maternal hospitalization. Mothers whose neonates were admitted to the NICU had a 30% higher likelihood of extended hospitalization compared to those whose newborns did not require NICU care (ARR = 1.30, 95% CI: 1.16–1.46). This finding is supported by studies conducted in Italy [11] and Sudan [9], which reported that maternal LoS was strongly influenced by neonatal complications requiring specialized care. The prolonged maternal stay in such cases is likely due to the need for continued breastfeeding, emotional support, and postpartum monitoring while the newborn receives intensive medical attention.

This study utilized a large sample size of post-cesarean cases and incorporated comprehensive data to enable robust comparison of variables associated with significant differences in LoS. Random sampling methods were employed to ensure representative selection. However, several limitations should be noted. The cross-sectional design prevents causal inference between identified factors and LoS. Recall bias may have influenced results due to reliance on self-reported maternal data. Moreover, unmeasured confounders could affect the observed associations, and institutional discharge policies or resource constraints may impact hospitalization duration independently of clinical needs.

## Conclusion

The results of this study demonstrated that in public general hospitals of the Sidama region, the median length of hospital stays for mothers after cesarean sections was 4 days. The length of hospital stays for mothers was determined by the age of the mother, neonatal intensive care unit admissions for newborns, mothers developing surgical site infections, and postoperative hemoglobin levels.

## Recommendations

General hospitals of Sidama Region should target high-risk mothers (older age, SSI, or anemia) with enhanced monitoring and wound care to accelerate recovery, while integrating NICU-maternity services and enforcing discharge criteria to maintain Ministry of Health of Ethiopia's recommendations without compromising safety.

## Supporting information

**S1 Data. Main LoS SPSS data.**
(SAV)

**S2 Data. Main LoS STATA data.**
(DTA)

## Acknowledgments

We would like to express our gratitude to Yirgalem Hospital Medical College for allowing to conduct this study. Special thanks to the chief executive directors of selected hospitals. We also extend our sincere appreciation to the study participants, dedicated data collectors, and supervisors for their invaluable contributions to this research.

## Author contributions

**Conceptualization:** Amelo Bolka, Zerihun Weldekidan.

**Data curation:** Amelo Bolka, Zerihun Weldekidan.

**Formal analysis:** Amelo Bolka, Zerihun Weldekidan.

**Funding acquisition:** Zerihun Weldekidan.

**Investigation:** Amelo Bolka, Zerihun Weldekidan.

**Methodology:** Amelo Bolka, Zerihun Weldekidan.

**Project administration:** Amelo Bolka.

**Resources:** Zerihun Weldekidan.

**Software:** Amelo Bolka.

**Supervision:** Amelo Bolka.

**Validation:** Amelo Bolka, Zerihun Weldekidan.

**Visualization:** Amelo Bolka, Zerihun Weldekidan.

**Writing – original draft:** Amelo Bolka, Zerihun Weldekidan.

**Writing – review & editing:** Amelo Bolka, Zerihun Weldekidan.

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
