## [Decision Letter · Decision Letter 0]

8 May 2025

PGPH-D-25-00744

Length of Hospital Stay and Its Associated Factors among Women Who Gave Birth by Cesarean Section in General Hospitals of Sidama Region, Southern Ethiopia

Dear Dr. Bolka,

Thank you for submitting your manuscript to PLOS Global Public Health. After careful consideration, we feel that it has merit but does not fully meet PLOS Global Public Health’s publication criteria as it currently stands. Therefore, we invite you to submit a revised version of the manuscript that addresses the points raised during the review process.

We look forward to receiving your revised manuscript.

Kind regards,

Shiyam Sunder, MBBS, MSc epidemiology

Academic Editor

Journal Requirements:

1. Please ensure that the Title in your manuscript file and the Title provided in your online submission form are the same.

2. Your manuscript is missing the following sections: Introduction. Please ensure these are present, and in the correct order, and that any references to subheadings in your main text are correct. An outline of the required sections can be consulted in our submission guidelines here:

https://journals.plos.org/globalpublichealth/s/submission-guidelines#loc-parts-of-a-submission

Additional Editor Comments (if provided):

Please address all comments of the reviewer.

Reviewers' comments:

Reviewer's Responses to Questions

**Comments to the Author**

1. Does this manuscript meet PLOS Global Public Health’s publication criteria?

Reviewer #1: Yes

Reviewer #2: Yes

2. Has the statistical analysis been performed appropriately and rigorously?

Reviewer #1: Yes

Reviewer #2: Yes

3. Have the authors made all data underlying the findings in their manuscript fully available (please refer to the Data Availability Statement at the start of the manuscript PDF file)?

Reviewer #1: Yes

Reviewer #2: Yes

4. Is the manuscript presented in an intelligible fashion and written in standard English?

Reviewer #1: Yes

Reviewer #2: Yes

Reviewer #1: A well written paper with a good methodology and results applicable to Ethiopian maternal health context. I have given some minor comments to further strengthen the paper:

Methods:

Some details of the hospitals where CS are conducted is needed in the methods section on study area and period. Details such as number of deliveries and C-section rates is required to get an overview of the case load in these ??six hospitals. It needs to be clear that the data is collected from tertiary hospitals where C section facility is available.

Results and Discussion:

If you can add a prevalence rate of C-sections in the Sidama region, as it would be a useful addition to the results section.

If possible, information on the cost of C-sections, including the hospital stay, is required since it can also influence the decision to discharge early.

Reviewer #2: Can author explain the following:

1. It is unclear why the sample size was calculated based on folate intake adequacy, as this variable was not assessed in the study (page 5).

2.Explain how systematic sampling was implemented (page 6).

3.Which obstetric complications were considered in the study? (page 6)

4. Could you please clarify what is meant by 'grand multipara? (Page 10, Table 2).

5.Which chronic medical conditions were included in the study? (Page 10, Table 2).

6. Kindly specify the units for low birth weight (e.g., less than ___ g and up to 2,500 g) and normal birth weight (Table 3).

7.Could you please clarify what is meant by low and normal APGAR scores? (Table 3).

8. Please describe the obstetric complications that were included in the study (Table 3).

9. Can author explain how has recall bias potentially influenced the outcomes reported in this study? (page 17)

10. I think, this should be the median length of hospital stay. (page 17, conclusion).

**Do you want your identity to be public for this peer review?** For information about this choice, including consent withdrawal, please see our Privacy Policy

Reviewer #1: No

Reviewer #2: No

---

## [Editor Report · Decision Letter 1]

15 May 2025

Length of Hospital Stay and Its Associated Factors among Women Who Gave Birth by Cesarean Section in General Hospitals of Sidama Region, Ethiopia

PGPH-D-25-00744R1

Dear Mr. Bolka,

We are pleased to inform you that your manuscript 'Length of Hospital Stay and Its Associated Factors among Women Who Gave Birth by Cesarean Section in General Hospitals of Sidama Region, Ethiopia' has been provisionally accepted for publication in PLOS Global Public Health.

Best regards,

Shiyam Sunder, MBBS, MSc epidemiology, PhD

Academic Editor